# Development, Validation, and Application of High-Performance Liquid Chromatography with Diode-Array Detection Method for Simultaneous Determination of Ginkgolic Acids and Ginkgols in *Ginkgo biloba*

**DOI:** 10.3390/foods13081250

**Published:** 2024-04-19

**Authors:** Isaac Duah Boateng, Fengnan Li, Xiao-Ming Yang

**Affiliations:** 1School of Food & Biological Engineering, Jiangsu University, Zhenjiang 212013, China; boatengisaacduah@gmail.com (I.D.B.); lifengnan0118@163.com (F.L.); 2Certified Group, 199 W Rhapsody Dr, San Antonio, TX 78216, USA

**Keywords:** *Ginkgo biloba*, *Ginkgo biloba* leaves, method validation, ginkgolic acids, ginkgols, ginkgo tea

## Abstract

*Ginkgo biloba* leaves (GBLs), which comprise many phytoconstituents, also contain a toxic substance named ginkgolic acid (GA). Our previous research showed that heating could decarboxylate and degrade GA into ginkgols with high levels of bioactivity. Several methods are available to measure GA in GBLs, but no analytical method has been developed to measure ginkgols and GA simultaneously. Hence, for the first time, an HPLC-DAD method was established to simultaneously determine GA and ginkgols using acetonitrile (0.01% trifluoroacetic acid, *v*/*v*) as mobile phase A and water (0.01% trifluoroacetic acid, *v*/*v*) as mobile phase B. The gradient elution conditions were: 0–30 min, 75–90% phase A; 30–35 min, 90–90% phase A; 35–36 min, 90–75% phase A; 36–46 min, 75–75% phase A. The detection wavelength of GA and ginkgol were 210 and 270 nm, respectively. The flow rate and injection volume were 1.0 mL/min and 50 μL, respectively. The linearity was excellent (*R*^2^ > 0.999), and the RSD of the precision, stability, and repeatability of the total ginkgols was 0.20%, 2.21%, and 2.45%, respectively, in six parallel determinations. The recoveries for the low, medium, and high groups were 96.58%, 97.67%, and 101.52%, respectively. The limit of detection of ginkgol C13:0, C15:1, and C17:1 was 0.61 ppm, 0.50 ppm, and 0.06 ppm, respectively. The limit of quantification of ginkgol C13:0, C15:1, and C17:1 was 2.01 ppm, 1.65 ppm, and 0.20 ppm, respectively. Finally, this method accurately measured the GA and ginkgol content in ginkgo leaves and ginkgo tea products (ginkgo black tea, ginkgo dark tea, ginkgo white tea, and ginkgo green tea), whereas principal component analysis (PCA) was performed to help visualize the association between GA and ginkgols and five different processing methods for GBLs. Thus, this research provides an efficient and accurate quantitative method for the subsequent detection of GA and ginkgols in ginkgo tea.

## 1. Introduction

*Ginkgo biloba* (GB) is widely recognized as a prominent medicinal plant due to its bioactive constituents, and the extracts derived from this botanical genus are frequently employed in the therapeutic management of many medical conditions, such as arteriosclerosis and rheumatism. The GB (Figure 1A) has been used for food and traditional herbal treatment in Japan, Korea, and China for centuries [1,2,3,4]. Numerous bioactive compounds, including flavone glycosides, ginkgolides, and bilobalides, have been extracted from *Ginkgo biloba* leaves (GBLs) (Figure 1B) and have demonstrated various physiological functions. These effects include mitigating anoxia and ischemia in the eye, brain, and heart, alleviating arteriosclerosis and rheumatism, suppressing platelet-activating factors, and regulating neurotransmitters [5,6,7]. Moreover, the global application of the extract of *Ginkgo biloba* (EGb) in the pharmaceutical, cosmetic, and functional food industries has been substantial, with sales >USD10 billion since 2017 [8]. 

Nonetheless, due to gingkolic acids (Figure 1C) being cytotoxic, genotoxic, mutagenic, neurotoxic, and allergenic, EGb utilization has been limited, as the United States, Chinese, and European Pharmacopoeias, in 2020, authorized that the concentration of gingkolic acids (GAs) in the GBE should be <5 µg/g [4,7,9]. According to our previous study [2], upon heating, GA undergoes decarboxylation, resulting in the formation of ginkgols (3-alkylphenols), which exhibit several pharmacological properties, including antibacterial, antioxidant, and anti-apoptotic actions [10]. These effects have been seen in vitro on tumor cells [3]. Moreover, in our previous study [2], it was observed that ginkgols (Figure 1D) had exceptional thermal stability and demonstrated anticancer effects when ginkgol C17:1, C15:1, and C13:0 were isolated. Since heating can decarboxylate and degrade GA into ginkgol, measuring ginkgol at the same time as GA would be more conducive to understanding the degradation of GA. 

Methods for detecting ginkgolic acids, including the UV, HPLC, and LC-MS methods, have been researched [11]. The detection of GA in EGb and GBLs in various pharmacopeias is mainly based on the HPLC method. In Jianbiao et al.’s [12] research, the authors found that the GA content detected via the European Pharmacopoeia version 7.0 method [13] was much higher than the method shown in the Chinese Pharmacopoeia 2010 version [14]. Therefore, the 2015 version of the Chinese Pharmacopoeia changed the mobile phase to 0.1% trifluoroacetic acid (TFA)–acetonitrile solution and 0.1% TFA-aqueous solution [15]. The primary differences between the Chinese and the European Pharmacopoeia version 7.0 detection method concern the chromatographic column, detection wavelength, and trifluoroacetic acid. Compared with the 2015 version of the Chinese Pharmacopoeia, the 2020 version [16] mainly strictly controls the GA content in EGb from <10 µg/g to <5 µg/g. However, the detection method has not been further updated. 

Since ginkgols naturally exist in GBLs and may also be generated through the degradation of GA during processing [2], measuring ginkgols at the same time as GA can effectively monitor the degradation of GA. Although Weihong et al. [17] and Liu [18] have developed a detection method for ginkgolic acids, the mobile phase is consistent with the 2010 version of the Chinese Pharmacopoeia method [14]. With the update to the Pharmacopoeia method, a new method for the simultaneous detection of ginkgolic acids and ginkgols should be developed. Nonetheless, there is no literature on a developed method that can simultaneously determine ginkgolic acids and ginkgols. Furthermore, there is no literature regarding the validation of this method and the application of the developed method on gingko leaves and gingko tea products. 

Therefore, this article establishes an accurate, reproducible, and precise HPLC method for simultaneously measuring ginkgols and ginkgolic acids and conducted methodological verification to detect ginkgols. Additionally, this method was employed to quantify the GA and ginkgols in various processed GBL teas and freeze-dried GBLs. Principal component analysis (PCA) was performed to visualize the association between GA and ginkgols and five different processing methods for GBLs. We believe that this method can produce commercial quality-control applications for GBL products.

## 2. Materials and Methods

### 2.1. Materials and Chemicals

Fresh GBLs were picked in Pizhou City, Jiangsu Province, on 29 May 2021. After selection, the GBLs were frozen at −20 °C for 24 h, freeze-dried at 50 °C (FreeZone Triad Cascade, Labconco Corporation, Kansas City, MO, USA) for 72 h, and ground into powder. The powdered GBLs were passed through a 40-mesh sieve and stored at −20 °C.

The standard mixture of GA homologs (their % of concentration were C13:0 (20.7%), C15:0 (3.3%), C15:1 (51.6%), and C17:1 (21.1%), and C17:2 (3.3%)) with a purity ≥98% was prepared in our lab based on our previous research [2]. The ginkgolic acids C13:0, C15:1, and C17:1 standard (purity ≥ 98%) were purchased from Shanghai Yuanye Company (Shanghai, China). The standards of ginkgols C13:0, C15:1, and C17:1 were synthesized by our laboratory [2], with purity >98%. Methanol and acetonitrile (HPLC grade) were purchased from Tedia Company (Shanghai, China), while the trifluoroacetic acid (HPLC grade) was bought from Shanghai McLean Reagent Company (Shanghai, China). The glacial acetic acid, petroleum ether, ethyl acetate, and other reagents and chemicals (analytical grade) were bought from Sinopharm Chemical Reagent Co., Ltd. (Shanghai, China).

### 2.2. Experimental Design

#### 2.2.1. Comparison of GA Detection Methods between European Pharmacopoeia 7.0 and Chinese Pharmacopoeia 2015 Edition 

European Pharmacopoeia 7.0 edition [13] and Chinese Pharmacopoeia 2015 edition [15] were applied to detect GA. The chromatographic column used for testing the Chinese Pharmacopoeia method was an Agilent Eclipse XDB-C18 column (250 × 4.6 mm, 5 μm), and the chromatographic conditions were as follows: mobile phase (A) 0.1% trifluoroacetic acid (TFA) acetonitrile solution (B) 0.1% TFA aqueous solution; gradient elution: mobile phase A: 0–30 min, 75–90%, 30–50 min: 90–90%, 50–51 min, 90–75%, 51–55 min, 75–75%. The detection wavelength, chromatographic packing, injection volume, flow rate, and column temperature were 310 nm, C18, 50 μL, 1.0 mL/min, and 35 °C, respectively.

Regarding the European Pharmacopoeia 7.0 edition [13], the column for the European Pharmacopoeia method was an Ultimate XB-C8 column ((250 × 4.6 mm, 5 μm) from Yuexu Technology Company (Shanghai, China) and the chromatographic conditions were as follows: mobile phase (A) 0.01% trifluoroacetic acid acetonitrile solution (B) 0.01% trifluoroacetic acid aqueous solution, gradient elution: mobile phase A: 0–30 min, 75–90%, 30–35 min: 90–90%, 35–36 min, 90–75%, 36–46 min, 75–75%. The detection wavelength, chromatographic packing, injection volume, flow rate, and column temperature were 210 nm, C8, 50 μL, 1.0 mL/min, and 35 °C, respectively.

#### 2.2.2. GA and Ginkgol Extraction from GBLs

Li et al.’s [7] protocol was applied for extraction. Freeze-dried GBL powder (0.5 g) was placed in a 50 mL centrifuge tube, 20 mL of methanol was added, and the total weight was recorded. The mixture was subjected to ultrasonication (40 °C, 100 W) for 45 min. The solution was cooled to room temperature, and methanol was added to recompensate the lost weight. The solution was centrifuged at 4669× *g* for 5 min at 4 °C. After diluting the supernatant 5×, the solution passed through a 0.45 μm filter. The solution was injected into the HPLC system (Agilent 1260, Agilent Corporation, Santa Clara, CA, USA) per the chromatographic conditions in Section 2.2.1. The peak area was recorded, and the contents of ginkgolic acid C13:0, C15:0, C15:1, C17:1, and C17:2 were determined. The total ginkgolic acid (TGA) was calculated as the sum of C13:0, C15:0, C15:1, C17:1, and C17:2 [7]. 

### 2.3. UV Absorption Spectrum for Ginkgolic Acids and Ginkgols

The GA and ginkgol standards were dissolved in acetonitrile solution to a concentration of 0.01 mg/ml and scanned within the 200–400 nm range to obtain the UV absorption, using a UV-visible spectrophotometer (UV-2450, Shimadzu Co., Ltd., Kyoto, Japan), to acquire a spectrum for GA and ginkgols [2].

### 2.4. Method Validation for GA and Ginkgols

The method of GA determination using European Pharmacopoeia version 7.0 has been validated [13]. Hence, in this study, we developed a reproducible method for detecting ginkgols in GBLs. Sample preparation was performed per the method stated in Section 2.2.2.

#### 2.4.1. Chromatographic Conditions and System Suitability Test

The chromatographic column was the Ultimate XB-C8 column (250 × 4.6 mm, 5 μm, Yuexu Technology Company, Shanghai, China). The HPLC conditions are shown in Section 2.2.1, and the ginkgols’ detection wavelength was set to 270 nm.

#### 2.4.2. Preparation of Test Samples and Reference Solution

Detailed preparation of test samples is shown in Section 2.2.2. Regarding reference solution preparation, 5.66 mg of ginkgol C13:0, 5.30 mg of ginkgol C15:1, and 13.88 mg of ginkgol C17:1 were diluted with 100 mL methanol (100%) to obtain 0.0566 mg/mL, 0.0530 mg/mL, and 0.1388 mg/mL of ginkgol C13: 0, C15:1, and ginkgol 17:1 stock solution, respectively. Afterwards, each stock solution was diluted with methanol into gradient concentrations of ginkgol C13:0 (0.10608 mg/mL), C15:1 (0.1435 mg/mL), and C17:1 (0.1864 mg/mL) standard solutions. 

#### 2.4.3. Linear Range, Limit of Detection (LOD), and Limit of Quantification (LOQ)

The prepared gradient concentration standard solution was injected per the chromatographic conditions in Section 2.2.1. The peak area was documented, and a standard curve was drawn with the peak area as the ordinate (Y) and the concentration as the abscissa (X), with a linear regression performed to estimate linearity. The LOD was estimated as 3× the signal-to-noise (S/N) ratio [19,20]. The LOQ was determined using the protocol of Ahuja and Dong [21]. 

#### 2.4.4. Precision, Stability, and Repeatability

For the precision test, following the procedure mentioned above (Section 2.4.2, reference solution), ginkgol stock solution (C13:0, C15:1, C17:1, mixed in a ratio of 3:4:3 *v*/*v*/*v*) was injected into the HPLC system per the chromatographic conditions in Section 2.4.1. The injection was performed 6 consecutive times, and the peak area was documented and substituted into the standard curve to estimate the ginkgol content and relative standard deviation (RSD) value.

For the stability test, the same test solution was injected once every 4 h based on the chromatographic conditions above. The injection was performed 6 consecutive times, and the peak area was noted and substituted into the standard curve to determine the ginkgol content and RSD value [20,22].

Finally, for the repeatability test, 6 parts of the test solution (3 different concentrations, each repeated 3 times; the test solution was repeated 6 times) were prepared, the samples were injected according to the chromatographic conditions, and the peak area was documented and substituted into the standard curve to compute the ginkgol content and RSD value [19].

#### 2.4.5. Recovery Rate

The recovery rateprocedure required 3 × 3 parts of the test solution (3 different concentrations, each repeated 3 times). The ginkgol standard solution (C13:0, C15:1, C17:1, in a ratio of 1:1:3 *v*/*v*/*v*) was added to the GBLs in proportions of 100%, 120%, and 150%, respectively. After diluting the solution 5 times, 9 portions of the test solution were prepared. The sample was injected according to the chromatographic conditions above (Section 2.2.1). The peak area was documented and substituted into the standard curve to determine the ginkgol content and RSD value [23].

### 2.5. Usage of This Proposed Method on Processed GBL Products

The proposed method was applied to ginkgo tea and freeze-dried ginkgo leaves to detect ginkgols and ginkgolic acids simultaneously. The GBLs (1 kg) were divided into 5 equal parts to make ginkgo tea [24]. 

I.To make ginkgo black tea (BT), the GBLs were kneaded in a kneading machine with a rotation speed of 55 rpm (voltage 380 V, motor speed 1400 rpm) for 10 min and fermented in an incubator at 40 °C and 80% humidity for 8 h.II.To make ginkgo dark tea (DT), the GBLs were dried in an oven at 150 °C for 5 min, kneaded at 55 rpm for 10 min, and fermented in an incubator at 40 °C and 80% humidity for 72 h.III.When making ginkgo white tea (WT), the GBLs were spread flat and withered at 20 °C for 72 h.IV.To make ginkgo green tea (GT), the ginkgo leaves were dried at 150 °C for 5 min and kneaded at 55 rpm for 10 min. All the above-mentioned semi-finished ginkgo teas were dried in an oven at 70 °C until their moisture content was approximately 7%.V.For the freeze-dried (FD) control group, GBLs were frozen at −20 °C for 24 h and freeze-dried at −50 °C for 72 h. They were then crushed, passed through a 40-mesh sieve, and stored at −20 °C.

In the extraction procedure for ginkgols and ginkgolic acids in ginkgo tea products, Li et al.’s [7] protocol was applied. The GBT powder (0.5 g) was placed in a 50 mL centrifuge tube. Methanol (20 mL) was added, and the total weight was recorded. This was subjected to ultrasonication (40 °C, 40 kHz, 100 W, pulse off-and-on time of 2 s:10 s) for 45 min, after which the solution was cooled and methanol was added to balance the lost weight. The solution was centrifuged at 4669× *g* for 5 min at 4 °C. After diluting the supernatant 5×, the solution passed through a 0.45 μm filter. 

### 2.6. Data Analysis

All experiments were performed in triplicate. Results are expressed as mean ± standard deviation. One-way analysis of variance (ANOVA) was performed on the ginkgol and ginkgolic acid homolog results using SPSS 26.0 software (IBM, Chicago, USA) and Tukey’s test. *p* < 0.05 showed that the difference was statistically significant. Origin 2019b software (OriginLab^®^, Northampton, MA, USA) was used for the principal component analysis of the ginkgo tea products and other graphical representations.

## 3. Results and Discussion

### 3.1. Comparison of GA Detection Methods between European Pharmacopoeia 7.0 and Chinese Pharmacopoeia 2015 Edition

The chromatograms of the GA homolog standards and samples measured via the GA detection methods are shown in Figure 2A–D. The chromatographic conditions of the two detection methods achieved good baseline separation for the five monomers of the homologous mixture of GA and each monomer of GA in GBLs (Figure 2A–D). Comparing the analysis time, the European Pharmacopoeia method completed peak elution within 25 min (Figure 2A,C), and the analysis time was shorter, while the Chinese Pharmacopoeia method required approximately 44 min to complete peak elution (Figure 2B,D). The peak width/peak height was evaluated [25]. These findings showed that the peak width/peak height of C13:0, C15:1, and C17:1 in the Chinese Pharmacopoeia method for the detection of GA were 0.2746/32.41, 0.2778/65.08, 0.5900/10.45 min/mAu, respectively. For the European Pharmacopoeia, the peak width/peak height of C13:0, 15:1, and C17:1 were 0.2514/209.3, 0.2554/454.9, and 0.2849/148.0 min/mAu, respectively.

Ginkgolic acid C13:0, C15:1, and C17:1 standard solutions (shown in Section 2.1) of different concentrations were injected, and the peak areas were measured to obtain the fitting of ginkgolic acid C13:0, C15:1, and C17:1 using the two detection methods. The standard curves are shown in Figure 2E–G. It can be seen that when the concentration of the standard GA was the same, the response value measured using the European Pharmacopoeia was much higher than the response value measured using the Chinese Pharmacopoeia. This is because 210 nm was the strong absorption of the phenyl ring of GA, and its intensity was much higher than 310 nm [23]. The absorption of the carboxyl group on the phenyl ring of GA was at 210 nm, and the detection sensitivity was higher. It is well known that the n---pi*band measured from COOH is of very low sensitivity. Therefore, the European Pharmacopoeia method is more suitable for detection at low concentrations than the Chinese Pharmacopoeia method. The European Pharmacopoeia method is more sensitive in determining the GA content in EGb and more suitable for detecting the GA limit standard in EGb. For non-polar compounds, such as ginkgo phenolic acids, both C18 and C8 columns are suitable for separation [26]; however, it is easier to use a C8 column than a C18 column [27]. It requires a small sample, there is no sample cleanup, the C8 RP-HPLC separation is fine, the chromatogram is clean, the sensitivity is sufficient, and it uses a readily available low-cost UV detector [26]. Therefore, the C8 column was decided upon for subsequent experiments.

The same GA in GBLs was measured using the Chinese Pharmacopoeia 2015 edition methods and the European Pharmacopoeia edition 7.0, as shown in Figure 2H. It was found that the C15:1 and C17:1 contents of GA and the total GA (TGA) measured using the European Pharmacopoeia method were significantly higher than those found using the Chinese Pharmacopoeia method. According to the test results, there were significant differences (*p* < 0.05) between the test results for the same sample (Figure 2H). The test results using the method specified in the European Pharmacopoeia were ~19.25% higher than those in the Chinese Pharmacopoeia. The main difference was due to the large difference in the results for ginkgolic acid C15:1 and C17:1 (Figure 2H), resulting in a high GA content measured in the European Pharmacopoeia. Comparing the two detection methods, the European Pharmacopoeia version 7.0 detection method had high sensitivity and a short peak time; the Chinese Pharmacopoeia method had slightly lower sensitivity and a longer peak time, especially since the retention time of C17:1 was close to 44 min (Figure 2B). Considering that the peak time was too long, this would have led to the peak broadening and affected the sensitivity [28]. Therefore, this study chose the method specified in the European Pharmacopoeia version 7.0 to develop a method to detect ginkgols and GA simultaneously. 

The standard mixture of GA homologous series and the homologous mixture of ginkgol was mixed and injected into the HPLC according to the European Pharmacopoeia 7.0 method. The chromatogram is shown in Figure 3A. As shown in Figure 3A, the retention times of the four monomers of ginkgolic acid, C13:0, C15:1, C15:0, and C17:1, were 17.46, 18.65, 23.281 and 24.51 min, respectively. The retention times of the four monomers of ginkgols, C15:1, C17:2, C15:0, and C17:1, are 21.07, 21.16, 22.29, and 27.27 min, respectively. However, ginkgol C13:0 and ginkgolic acid C17:2 overlapped (Figure 3A) and were not separated. 

Further comparisons were made between the chromatograms of ginkgol C13:0 injected alone (Figure 3B) and the GA mixture (Figure 3C). It was found that ginkgol C13:0 (Rt = 19.911 min, Figure 3B) and ginkgolic acid C17:2 (Rt = 19.979 min, Figure 3C) were very close; that is, the peak area at the retention time of 19.9 min may be the sum of the peak areas of ginkgol C13:0 and ginkgolic acid C17:2. The detection of other monomers of GA was not affected. The proportion of ginkgolic acid C17:2 in GA homologs was very small, and the content in GBLs was also very low (almost negligible). Neither the Chinese Pharmacopoeia [15] nor the European Pharmacopoeia [13] include ginkgolic acid C17:2 in the measurement of total GA. Ginkgol C13:0 is the main component of ginkgol homologs and cannot be ignored [3]. Therefore, we researched the absorption spectra of GA and ginkgols to select the wavelength for detecting ginkgol.

### 3.2. Selection of Wavelength for Detecting Ginkgol and GA

Figure 3D–G show the UV absorption spectra of GA and ginkgol in various solutions. As can be seen from Figure 3F, using acetonitrile solution, the two maximum absorption peaks of GA were located at 219 nm and 312 nm and the maximum UV absorption peaks of ginkgol were located at 206 nm and 272 nm (Figure 3G). Ginkgols had a maximum absorption of ~272 nm, while GA had almost no absorption at 268 nm. The UV scan of ginkgol and GA in the mobile phase of European Pharmacopoeia version 7.0 is shown in Figure 3D,E. Ginkgol had medium to strong absorption at 274 nm (Figure 3D), while GA had almost no absorption at 274 nm (Figure 3E). Therefore, based on detecting GA at 210 nm, ginkgol could be detected at 270 nm to avoid ginkgolic acid C17:2 influencing the detection of ginkgol C13:0. 

To verify the feasibility of this idea, the GA homologous mixed standard was injected and monitored at wavelengths of 210 nm, 270 nm, and 310 nm (Appendix A). Appendix A are the chromatograms of five homologs of ginkgolic acid at 310 nm, 270 nm, and 210 nm, respectively.

Appendix A show five sharp, symmetrical absorption peaks for GA, with consistent retention times. Since the benzene ring of ginkgolic acid has strong absorption at 210 nm, its absorption intensity is much higher than that of the carboxyl group on the benzene ring at 310 nm [23]. Therefore, the response value in Appendix A is ~10× that in Appendix A (regarding peak height). At the same time, in the 270 nm chromatogram of the GA mixed standard (Appendix A), except for a small impurity peak at 17.416 min (Rt of ginkgolic acid C13:0 = 17.42 min), no other peaks appeared. Ginkgol C13:0 had the earliest peak among the five ginkgols (Rt = 19.91 min). The retention times of the remaining four ginkgol monomers were 21.07, 21.16, 22.29, and 27.27 min, respectively. Therefore, the detection wavelength of ginkgol was changed and set at 270 nm, as there would be no interference from GA.

Ginkgol C13:0, C15:1, and C17:1 standards and ginkgolic acid C13:0, C15:1, C15:0, and C17:1 standards were mixed and injected, and their chromatograms at 270 nm and 210 nm were attained (Figure 4A,B). The chromatogram of ginkgol C13:0, C15:1, and C17:1 at 270 nm is shown in Figure 4A, and no interference from GA is seen. Figure 4B shows the chromatograms of ginkgol C13:0, C15:1, and C17:1 and ginkgolic acid C13:0, C15:1 and C17:1 at 210 nm. The baseline between the monomers of ginkgol and GA is complete. 

### 3.3. Methodological Validation

#### 3.3.1. Linear Range, Limit of Detection, and Limit of Quantification

To determine the method’s accuracy, standard calibration curves were determined by injecting standard solutions at six concentration levels [23]. A standard curve was drawn, and the linear regression equation for ginkgol is shown in Appendix A. The linearity of the method pertains to the extent of analyte concentration within which the method is applicable [29]. The standard curves of each ginkgol standard product showed an excellent linear relationship with an *R*^2^ < 0.999 and a good linearity at a similar concentration range [30]. The limit of detection (LOD), which refers to the minimum concentration that can be reliably quantified using a certain analytical method [30], was determined (Appendix A). The findings showed an LOD of 0.61, 0.5, and 0.06 ppm for C13:0, C15:1, and C17:1, respectively. The limit of quantification (LOQ) stands for the smallest amount or the lowest concentration of a substance that can be determined through a given analytical procedure with established accuracy, precision, and uncertainty [21]. The LOQ of ginkgol C13:0, C15:1, and C17:1 was 2.01 ppm, 1.65 ppm, and 0.20 ppm, respectively.

#### 3.3.2. Precision and Repeatability Experiments

The method’s precision was determined by injecting ginkgol standard stock solution (C13:0, C15:1, C17:1, 3:4:3 mixture) into the HPLC system using the above chromatographic conditions for six consecutive injections. As shown in Table 1, the RSD of the measured ginkgol C13:0, C15:1, and C17:1 standards was 0.27%, 0.33%, and 0.28%, respectively. The RSD of the total ginkgol content was 0.20%. The RSD values met the acceptable threshold, indicating that the approach demonstrated a rather high level of precision [23,31]. 

Repeatability experiments were conducted by taking six test samples and injecting them into the HPLC system per the above chromatographic conditions. As shown in Table 1**,** the RSDs of the measured ginkgol C13:0, C15:1, and C17:1 were 3.76%, 2.58%, and 2.40%, respectively. The total ginkols’ RSD was 2.45%, and the method stability was high. The RSD values were within the acceptable criteria and indicated good method [23,31]. 

#### 3.3.3. Stability Experiment

The same test solution was injected once every 4 h into the HPLC system, per the above chromatographic conditions, for six consecutive injections. As shown in Table 1, the RSD of the measured ginkgol C13:0, C15:1, and C17:1 was 3.06%, 2.93%, and 3.73%, respectively. The RSD of the total ginkgol content was 2.21%. The RSD values met the acceptable threshold, indicating that the approach demonstrated a rather good level of stability [23,31].

#### 3.3.4. Recovery

An experiment was conducted to validate the accuracy and precision of the suggested approach for detecting ginkgols in *Ginkgo biloba* leaves [30]. Samples of the below 9 test solutions were injected into the HPLC, the data were recorded, and the results were calculated according to the standard curve. The recovery rates of ginkgol C13:0, C15:1, and C17:1 are shown in Table 2. The recoveries of the measured ginkgol C13:0 content (low, medium, and high sample groups) were 95.82%, 96.41%, and 95.60%, respectively, and the RSDs were 3.93%, 2.90%, and 3.72%, respectively. The recoveries of the measured ginkgol C15:1 content (low, medium, and high sample groups) were 95.82%, 96.41%, and 93.63%, respectively, and the RSDs were 3.24%, 2.86%, and 2.97%, respectively. The recoveries of the measured ginkgol C17:1 content (low, medium, and high loading groups) were 97.45%, 97.49%, and 103.94%, respectively, and the RSDs were 1.81%, 3.44%, and 1.94%, respectively. The acceptable recovery range for most residue analysis technique guidelines is typically between 70% and 120% [32]. Thus, our results (Table 2) show that our recovery range (~94 to 98%) is acceptable.

Finally, ginkgol C13:0, C15:1, and C17:1 were added to obtain the total ginkgol content. The calculations of the total ginkgo acid content of each sample are shown in Table 2. The recovery rate of ginkgol (low, medium, and high sample addition groups) was 96.58%, 97.67%, and 101.52%, respectively, and the RSDs were 1.40%, 2.46%, and 1.56%, respectively. The RSD values met the acceptable threshold, indicating that the approach demonstrated a rather high level of precision [31]. 

### 3.4. Application of the Developed Method to Investigate Ginkgols and GA in Processed GBLs

The developed method was employed to quantify the GA and ginkgols in various processed GBL teas and freeze-dried GBLs. The HPLC chromatogram is found in Figure 5, showing a baseline separation. The contents of GA homologs C13:0, C15:1, and C17:1 account for more than 90% of the total ginkgolic acid [24], and our results (Figure 4C,D) attest to that. It can be seen from Figure 4D that the total ginkgolic acid content of dark tea (30.07 mg/g) was significantly lower than those of the other three ginkgo teas and freeze-dried GBLs; the ginkgolic acid content of white tea was the highest among all tested samples, which may be due to withering [24]. The process leads to the accumulation of intracellular oxygen free radicals and the loss of selective permeability in the cell membrane [33]. Eventually, the cell contents burst, and ginkgolic acid flows out of the cells. After the tea is made, the ginkgolic acid content is relatively high. It should be noted that GA in all the GBTs was >5 ppm, thus requiring combined multiple physical processing technologies in GBT processing; hybrid processing is increasing [34]. In a study by Li et al. [7], the pulsed-light irradiation of GBLs improved the retention of the main bioactive compounds and the inactivation of polyphenol oxidase (PPO) and polyphenol peroxidase (POD) in GBLs compared to the control. Hence, combining new physical processing technology with fermentation technology used for GBTs may provide a way to reduce the GA content and retain bioactive ingredients, such as ginkgo flavonoids and lactones, while processing GBTs.

Three main types of ginkgols (ginkgol C13:0, C15:1, and C17:1) were measured in freeze-dried GBLs and ginkgo tea. Figure 4C shows the measurement results for three kinds of ginkgols and the total ginkgol content in ginkgo tea. Figure 4C shows that the total ginkgol content in dark tea was the highest and was significantly different from the content in freeze-dried GBLs and the other three ginkgo teas (*p* < 0.05). The total ginkgol content of dark tea was ~3.84 mg/g dt, which was ~2.76 mg higher than that of freeze-dried GBLs (Figure 4C), while the ginkgolic acid of dark tea was ~2.80 mg lower than that of freeze-dried GBLs (Figure 4D). The increased content of ginkgol is similar to the content degraded by ginkgolic acid. It is speculated that the combination of fixation [2] and fermentation [35] during dark tea production causes ginkgolic acid degradation, thus increasing ginkgol content. No similar phenomenon was found in other ginkgo teas, such as green and lightly fermented black tea. Compared with freeze-dried GBLs (Figure 4C,D), their ginkgolic acid content decreased and their ginkgol content increased slightly, but there was no significant difference (*p* > 0.05). The contents of ginkgolic acid and ginkgol in withered white tea were marginally higher than those in freeze-dried GBLs, possibly related to the outflow of cell contents caused by withering [24].

### 3.5. Principal Component Analysis (PCA)

PCA was performed (using the raw values of the ginkgolic acids and ginkgol homologs in Figure 4C,D) to visualize the association between GA and ginkgols and five different processing methods for GBLs (Figure 4E). PCA is an unsupervised statistical approach that uses the reduction of the dimensions of the principal components to analyze variations across sample groups [7]. Figure 4E shows the biplot of the first two principal components, PC1 and PC2, representing 92.79% of the total variability. A cumulative principal component greater than 85% is high enough to explain the total variance in the dataset [36,37]. Thus, our selection of PC1-2 is justifiable for accurately differentiating bioactive compounds among groups, and there is no need to perform orthogonal partial least squares discriminant analysis (OPLS-DA), used to mitigate environmental and systematic errors. PC1 (69.87%) describes the most sample variation. It also means that PC1 explained most related parameters, whereas PC2 and PC3 described lesser correlated parameters. In the biplot (Figure 4E), black tea (BT), white tea (GT), and freeze-dried GBLs appeared on PC1’s positive side, while dark tea (DT) was on the negative side of PC1. This means that by using our developed method, we can classify ginkgo tea products based on the content of ginkgolic acid and ginkgols. In Figure 4B, FD shares the same characteristics as BT and GT, meaning that their processing methods have similar ginkgols and ginkgolic acid content. Furthermore, it was observed that gingkols were on the negative side of PC1, whereas ginkgolic acids were on the positive side of PC1. This proves that the increased content of ginkgol obtained by decarboxylating ginkgolic acids to ginkgols (3-alkylphenols) is due to processing treatments and aligns with the findings of Yang et al. [2]. 

The eigenvalues were calculated (Appendix A) to understand which processing method and compound (various ginkgolic acids and ginkgols) contributed significantly to the PCA. The findings (Appendix A) showed that PC1 was contributed considerably by GA C13:0 (0.4222) and ginkgol C17:1 (−0.3812), whereas PC2 was significantly contributed by ginkgol C13:0 (0.5181) and ginkgolic acid C15:1 (0.4168) based on their Eigenvector coefficients, and these align with the biplot (Figure 4E). Pearson’s correlations were used to establish and characterize the relationships between the ginkgolic acid and ginkgols, and the results (Appendix A) attested to the PCA findings (Figure 4E), thus indicating that the processing method affected ginkgolic acids and ginkgols. 

## 4. Conclusions

This research used the methods of the Chinese Pharmacopoeia 2015 edition and the European Pharmacopoeia edition 7.0 to measure and compare the ginkgolic acid contents in GBLs. It was found that the accuracy of the European Pharmacopoeia method was ~19.25% higher than the Chinese Pharmacopoeia method and its sensitivity was higher than that of the Chinese Pharmacopeia. An HPLC-DAD method was established to simultaneously determine GA and ginkgols.. The detection wavelengths of GA and ginkgol were 210 and 270 nm, respectively. The flow rate and injection volume were 1.0 mL/min and 50 μL, respectively. The linearity had *R*^2^ > 0.999, and the RSDs for the precision, stability, and repeatability of total ginkgols were 0.20%, 2.21%, and 2.45%, respectively, in six parallel determinations. The recoveries of low, medium, and high groups were 96.58%, 97.67%, and 101.52%, respectively, and were within the acceptable criteria. The LOD of ginkgol C13:0, C15:1, and C17:1 were 0.61 ppm, 0.50 ppm, and 0.06 ppm, respectively, whereas the LOQ of ginkgol C13:0, C15:1, and C17:1 were 2.01 ppm, 1.65 ppm, and 0.20 ppm, respectively. Finally, this method was able to accurately measure the GA and ginkgol content in five different ginkgo products that can be used for the simultaneous determination of the GA and ginkgol contents in various processed ginkgo leaves (freeze-dried ginkgo leaves, ginkgo black tea, ginkgo dark tea, ginkgo white tea, and ginkgo green tea). Future studies could evaluate the use of the European Pharmacopoeia column with the Chinese Pharmacopoeia eluent and gradient, and vice versa.

## Figures and Tables

**Figure 1 foods-13-01250-f001:**
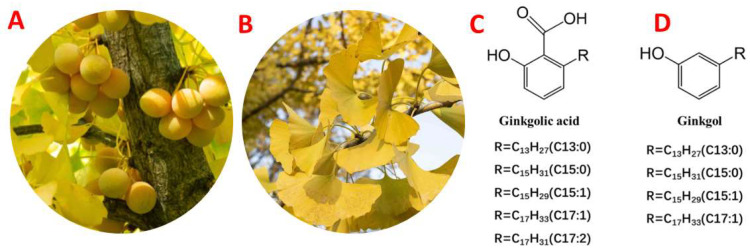
*Ginkgo biloba* tree (**A**), leaves (**B**), and structures of ginkgolic acids (**C**) and ginkgols (**D**).

**Figure 2 foods-13-01250-f002:**
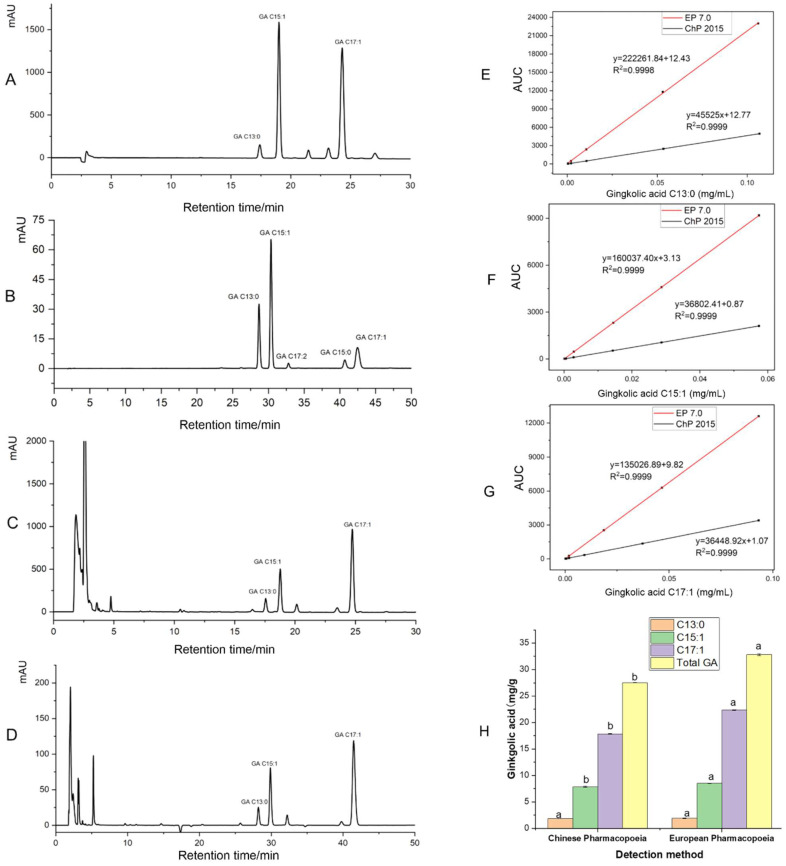
(**A**–**D**) Chromatogram of ginkgolic acid standard (**A**) and freeze-dried ginkgo leaf (**C**) using European Pharmacopoeia method, ginkgolic acid standard (**B**) and freeze-dried ginkgo leaf (**D**) using Chinese Pharmacopoeia method. (**E**–**G**) Chinese Pharmacopoeia and European Pharmacopoeia methods to detect ginkgolic acid C13:0 (**E**), C15:1 (**F**), and C17:1 (**G**) standard curves. (**H**) The content of ginkgolic acid in ginkgo leaves was detected via different detection methods. **Note:** different letters indicate significant differences in the same compound, *p* < 0.05, EP 7.0 using Tukey test. EP 7.0, European Pharmacopoeia 7.0 version method; ChP 2015, Chinese Pharmacopoeia 2015 version method.

**Figure 3 foods-13-01250-f003:**
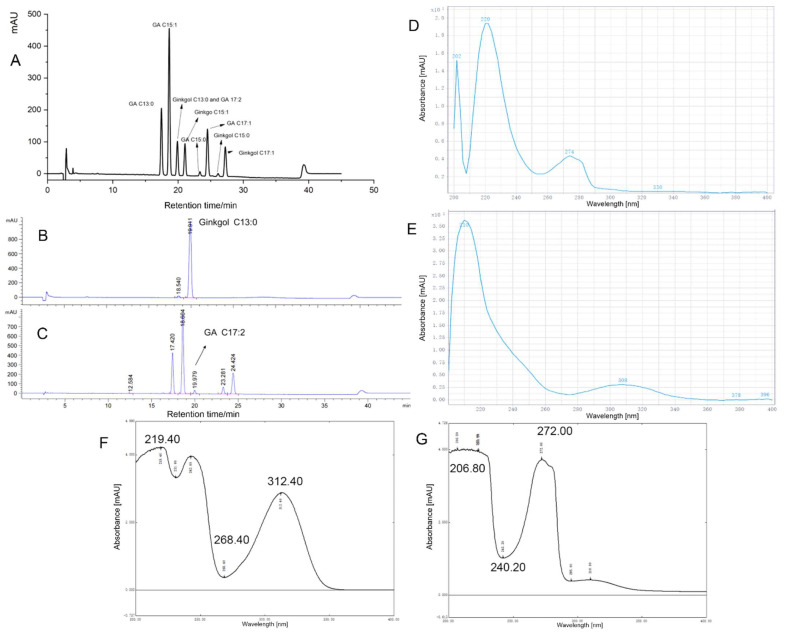
(**A**) Chromatogram of the European Pharmacopoeia method for standard ginkgolic acid mixed with ginkgol. (**B**,**C**) Chromatogram of ginkgol C13:0. (**B**) Ginkgolic acid mixture (**C**) detected at 210 nm using the European Pharmacopoeia method. (**D**,**E**) The UV spectra of ginkgol (**D**) and ginkgolic acid (**E**) in 0.01% trifluoroacetic acid acetonitrile solution–0.01% trifluoroacetic acid aqueous solution. (**F**,**G**) UV spectrum of ginkgolic acid (**F**) and ginkgol (**G**) in acetonitrile solution.

**Figure 4 foods-13-01250-f004:**
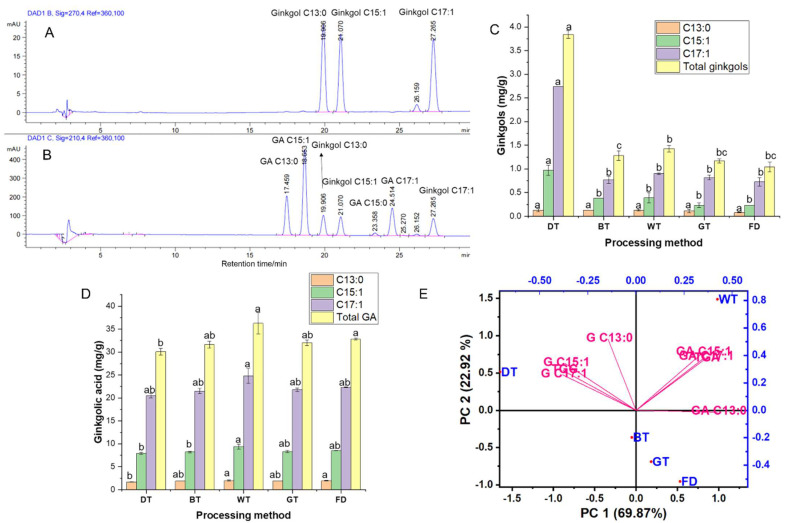
(**A**,**B**) Chromatogram of ginkgol (**A**) and the mixture of ginkgol and ginkgolic acid (**B**) detected using the European Pharmacopoeia method. (**C**) Ginkgol and ginkgolic acid (**D**) content in ginkgo leaf tea and freeze-dried ginkgo leaves. (**E**) Three-dimensional biplot from principal component analysis. **Note.** Different letters indicate significant differences in the same compound, *p* < 0.05 using the Tukey test. BT, black tea, WT, white tea, GT, green tea, DT, dark tea, and FD, freeze-dried GBLs.

**Figure 5 foods-13-01250-f005:**
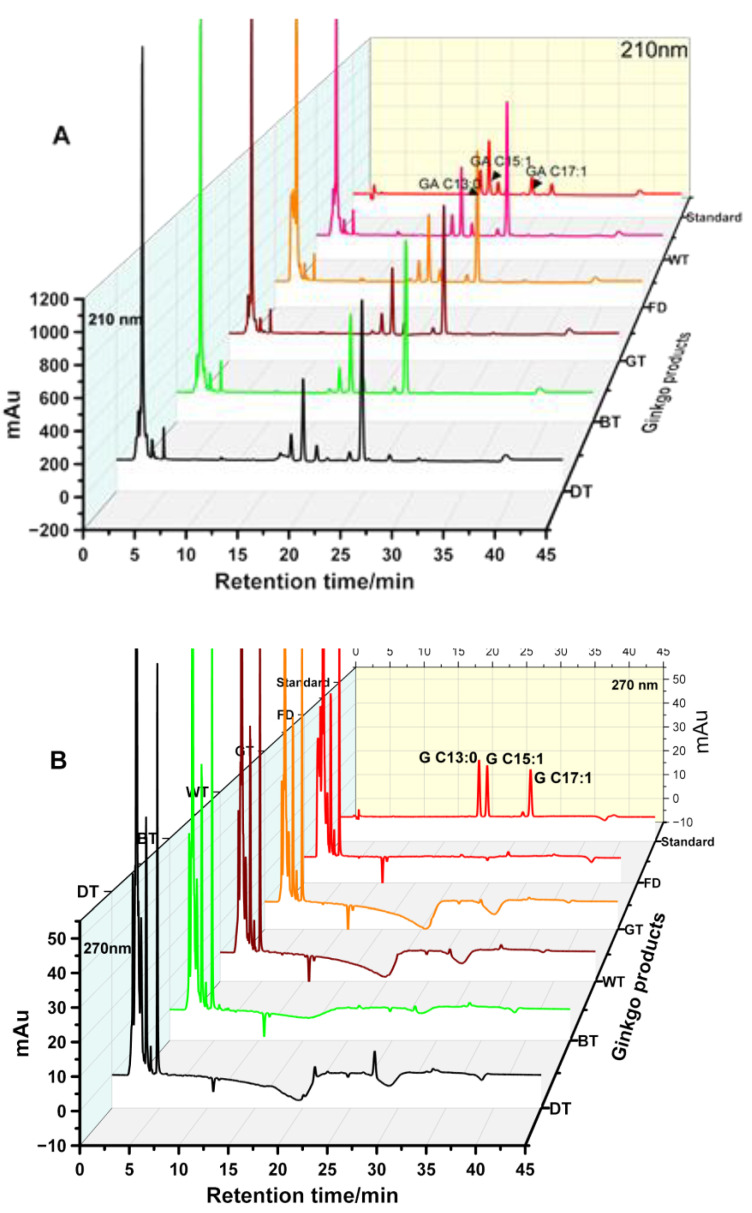
HPLC chromatogram of (**A**) ginkgolic acids and (**B**) ginkgols in various ginkgo products.

**Table 1 foods-13-01250-t001:** Method validation for precision, stability, and repeatability of test results.

Standard Solution Precision Test	Standard Solution	Ginkgol C13:0 (mg/mL)	Ginkgol C15:1 (mg/mL)	Ginkgol C17:1 (mg/mL)	Total Ginkgol (mg/mL)
	1	0.0244	0.0295	0.0430	0.0969
	2	0.0245	0.0297	0.0431	0.0972
	3	0.0245	0.0297	0.0431	0.0973
	4	0.0246	0.0297	0.0432	0.0975
	5	0.0245	0.0295	0.0433	0.0974
	6	0.0246	0.0298	0.0430	0.0974
	Average value	0.0245	0.0297	0.0431	0.0973
	RSD	0.27%	0.33%	0.29%	0.20%
Sample solution stability test	Time (h)	Ginkgol C13:0 (mg/g)	Ginkgol C15:1 (mg/g)	Ginkgol C17:1 (mg/g)	Total ginkgol (mg/g)
	0	0.0259	0.3344	0.6511	1.0110
	4	0.0276	0.3351	0.6328	0.9981
	8	0.0278	0.3542	0.6570	1.0389
	12	0.0265	0.3540	0.6518	1.0324
	16	0.0285	0.3572	0.6390	1.0247
	20	0.0275	0.3591	0.5866	0.9731
	Average value	0.0273	0.3490	0.6364	1.0130
	RSD	3.06%	2.93%	3.73%	2.21%
Sample solution repeatability test	Sample	Ginkgol C13:0 (mg/g)	Ginkgol C15:1 (mg/g)	Ginkgol C17:1 (mg/g)	Total ginkgol (mg/g)
	1	0.0272	0.3572	0.6390	1.0247
	2	0.0264	0.3598	0.6459	1.0321
	3	0.0249	0.3518	0.6505	1.0272
	4	0.0244	0.3477	0.6252	0.9973
	5	0.0254	0.3367	0.6115	0.9735
	6	0.0251	0.3370	0.6138	0.9736
	Average value	0.0256	0.3484	0.6310	1.0047
	RSD	3.76%	2.58%	2.40%	2.45%

**Table 2 foods-13-01250-t002:** The recovery rate experiment results for ginkgol C13:0, C15:1, C17:1 in *Ginkgo biloba* leaves.

C13:0	Sample	Sample Amount (g)	C13:0 (mg)	Added Amount (mg)	Theoretical Value (mg)	Actual Value (mg)	Total Recovery Rate (%)	Average Value (%)	RSD (%)
	1	0.5014	0.0147	0.0556	0.0703	0.0643	91.50	95.82	3.93
	2	0.4985	0.0146	0.0556	0.0702	0.0669	95.29		
	3	0.5015	0.0147	0.0556	0.0703	0.0708	100.67		
	4	0.5046	0.0148	0.0667	0.0815	0.0798	97.91	96.41	2.90
	5	0.5010	0.0147	0.0667	0.0814	0.0804	98.83		
	6	0.5011	0.0147	0.0667	0.0814	0.0753	92.48		
	7	0.5018	0.0147	0.0834	0.0981	0.0927	94.49	95.60	3.72
	8	0.5020	0.0147	0.0834	0.0981	0.0985	100.40		
	9	0.4999	0.0146	0.0834	0.0980	0.0901	91.91		
C15:1	Sample	Sample amount (g)	C15:1 (mg)	Added amount (mg)	Theoretical value (mg)	Actual value (mg)	Total recovery (%)	Average value (%)	RSD (%)
	1	0.5014	0.1756	0.0795	0.2551	0.2501	98.01	94.52	3.24
	2	0.4985	0.1746	0.0795	0.2541	0.2414	95.01		
	3	0.5015	0.1757	0.0795	0.2552	0.2311	90.56		
	4	0.5046	0.1768	0.0954	0.2722	0.2621	96.32	98.52	2.86
	5	0.5010	0.1755	0.0954	0.2709	0.2777	102.50		
	6	0.5011	0.1755	0.0954	0.2709	0.2621	96.75		
	7	0.5018	0.1758	0.1193	0.2950	0.2966	100.55	96.61	2.97
	8	0.5020	0.1759	0.1193	0.2951	0.2768	93.80		
	9	0.4999	0.1751	0.1193	0.2944	0.2811	95.50		
C17:1	Sample	Sample amount (g)	C17:1 (mg)	Added amount (m)	Theoretical value (mg)	Actual value (mg)	Total recovery (%)	Average value (%)	RSD (%)
	1	0.5014	0.3153	0.3470	0.6623	0.6371	96.21	97.45	1.81
	2	0.4985	0.3134	0.3470	0.6604	0.6601	99.95		
	3	0.5015	0.3153	0.3470	0.6623	0.6371	96.20		
	4	0.5046	0.3173	0.4164	0.7337	0.6831	93.11	97.49	3.44
	5	0.5010	0.3150	0.4164	0.7314	0.7176	98.11		
	6	0.5011	0.3151	0.4164	0.7315	0.7406	101.25		
	7	0.5018	0.3155	0.5205	0.8360	0.8683	103.86	103.94	1.94
	8	0.5020	0.3156	0.5205	0.8361	0.8900	106.44		
	9	0.4999	0.3143	0.5205	0.8348	0.8475	101.52		
Total ginkgol	Sample	Sample amount (g)	Total ginkgol (mg)	Added amount (mg)	Theoretical sum (mg)	Actual sum (mg)	Total recovery rate (%)	Average value (%)	RSD (%)
	1	0.5014	0.5056	0.4821	0.9877	0.9515	96.34	96.58	1.40
	2	0.4985	0.5027	0.4821	0.9848	0.9685	98.34		
	3	0.5015	0.5057	0.4821	0.9878	0.9390	95.06		
	4	0.5046	0.5088	0.5785	1.0873	1.0250	94.27	97.67	2.46
	5	0.5010	0.5052	0.5785	1.0837	1.0757	99.26		
	6	0.5011	0.5053	0.5785	1.0838	1.0780	99.46		
	7	0.5018	0.5060	0.7232	1.2291	1.2518	102.32	101.52	1.56
	8	0.5020	0.5062	0.7232	1.2293	1.2653	102.93		
	9	0.4999	0.5041	0.7232	1.2272	1.2187	99.31		

## Data Availability

The original contributions presented in the study are included in the article/Appendix A, further inquiries can be directed to the corresponding author.

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
