# Peer review of "Development, Validation, and Application of High-Performance Liquid Chromatography with Diode-Array Detection Method for Simultaneous Determination of Ginkgolic Acids and Ginkgols in Ginkgo biloba"

_foods, 2024, doi:10.3390/foods13081250_

Round 1

Reviewer 1 Report

Comments and Suggestions for Authors

 Based on a partial Google Scholar search, the separaton of both ginkgolic acids and ginkgols, particularly a total of seven components, has not been reported before, as asserted by the authors. The analytical figures of merit are good. 

Comments on the Quality of English Language

The experimental section 2.4 has too many sub-headings. The numbering in section 2.5 is unnecessary. All sentences should start with a noun not a number.

Author Response

find attached the response to comments

Reviewer 2 Report

Comments and Suggestions for Authors

The paper was well written and conducted, I have found very few mistakes, and I have very few suggestions

 1.       Line 28 - Change ".... and component analysis (PCA) was performed ..." BY " ...whereas, component analysis (PCA) was performed ..."

2.       Line 58 -  in vitro should be Italic

3. Lines 190 to 203: It would be better use roman nunbers I to V, or letters, instead indo-arabic numbers (1 to 5).

3.3.2. Precision Experiment AND 3.3.4. Repeatability Experiments SHOULD BE CONNECTED

5

Comments on the Quality of English Language

The quality of english is good

Author Response

find attached the response to comments

Reviewer 3 Report

Comments and Suggestions for Authors

Comments on the Quality of English Language

Author Response

find attached the response to comments

Reviewer 4 Report

Comments and Suggestions for Authors

The paper aims to compare two Pharmacopeial methods used to evaluate ginkols and ginkolic acids in Ginkgo biloba leaves. While the study is interesting, there are some issues that need to be addressed:

  1. - Explain what GBL stands for when it is first mentioned in the text (in the Introduction).

  2. - Clarify the meaning of EGb.

  3. - In line 106, provide an explanation for why a temperature range of 60-90°C is provided.

  4. - Include information about tea preparation for HPLC analysis, as currently only different treatments are described.

  5. - Explain in the text how the total ginkolic acids are calculated. Is it the sum of C13:0, C15:1, and C17:1, or does it also include C15:0 and C17:2?

  6. - It would be valuable to evaluate the European Pharmacopoeia column with Chinese Pharmacopoeia eluent and gradient, and vice versa.

  7. - Provide recommendations for tea preparation or usage, as the detected amounts exceed recommended limits.

  8. - check journal requirements for references; some page numbers are missed; Latin names must be in italics.

Author Response

find attached the response to comments

Reviewer 5 Report

Comments and Suggestions for Authors

The manuscript entitled ‘Development, Validation, and Application of HPLC-DAD 2 Method for Simultaneous Determination of Ginkgolic Acids  and Ginkgols in ginkgo biloba’ by Boateng et al. to this journal is describing a new method for the simultaneous determination  and quantification of two important  related family of natural compounds the ginkgolic acids with the corresponding ginkgols.

The new methods derived from the actual European and Chinese pharmacopeia which describe the chromatographic procedure for the quantification of these two families of compounds separately, per forming a single analysis. The proposed methods have been subjected to the request normally required for the validation of a new methods.

I found the manuscript interesting and suitable for this journal after some amendments.

Line 52. Is reported EGb without explaining the meaning.

Lines 98-99. Explain the value of the percentage, by weight, mole etc.

Line 113 and 120. The two methods use two different columns, one is a C18 the other is a C8. I think that a brief comment on that should be added to the text.

Line 158. Here are described the linearity and the limit of detection and that is fine, but along the manuscript is not described at all about the limit of quantification (LOQ), that must be part of this study. The Authors must add that in the material and methods section and the relative comment in the text.

Line 204 In the manuscript is reported that the results were analyzed in a statistical manner, but along the text there is practically no or tiny sign of that, I strongly suggest adding all the ANOVA and post hoc analysis  in the supplementary information.

Line 321 Is not clear where the Authors took the numbers used to add at the PCA data for FD, GT, BT etc., give a short explanation in the text and add them in the supplementary information.

Line 356 <Producibility>, I’m not fully convinced that this word is put in the right context.

Line 360 <Above>, I think that should be -below-.

Lines 412 to 423 Viewing the three-dimensional biplot is not easy for a reader, given all the numbers in the supplementary information it will be easy to reconstruct the 3D figure in any specialized software. I ask the author if it will be appropriate to represent only PC1 and PC2 which have a variability greater than 90% with a 2D biplot?

Line 536 Three chromatograms are represented but only two are listed in the figure caption. Double check the wavelengths.

I was surprised not to see the complete chromatograms of the five types of ginkgo tea and not discuss them in terms of the targeted analytes and all the other associated molecules and whether they caused problems in analyte discrimination. I highly recommend adding all these parts to the manuscript in the appropriate sections.

Author Response

find attached the response to comments

Reviewer 6 Report

Comments and Suggestions for Authors

The comments for Authors of the manuscript entitle: Development, validation, and application of HPLC-DAD method for simultaneous determination of ginkgolic acids and ginkgols in Ginkgo biloba

The manuscript presents the application of the HPLC method for the simultaneous determination of ginkgolic acid and ginkgols based on the methods in the pharmacopoeia. The Authors demonstrated the suitability of the developed method for the determination of ginkgolic acid and ginkgols in ginco tea products. The method has undergone a validation process. The manuscript is a concise, well written text describing the analyses carried out. I belive the presented manuscript is suitable for publication in the journal Food, but in my opinion it needs some minor corrections.

In my opinion the biggest deficiency of the presented manuscript is the Materials and Methods section. The description contains too many subsections and references to sections 2.2.2 and 2.2.1. The entire description should be rewritten. There is no need to constantly refer to the mentioned sections. The validation process should be presented in one chapter.

Other comments:

Line 46: GBE is not in the list of abbreviations. Is this the same as EGb? EGb is also not in the list of abbreviations.

Line 52: EGb GBE placed together, Should this be the case since they mean the same thing? (In my opinion they mean the same thing)

Line 58 in vitro – use italics

Figure 2. In my opinion better is to present chromatograms considering Eur. Pharmacopoeia together and these considering Chin. Pharmacopoeia also together. Therefore, it would be better to substitute places (B) instead of (C).

Figure 3. there is no scale (retention time) at Fig. 3B. In Fig. 3 D, E, F and G, the scale is not very clear.

The caption of Figure 3, line 271 is: mixed with ginkgolic acid; shouldn't it be a GA mixture?

In my opinion Figure S1 should be placed in the main text.

Figure 4: I wonder why there is no GA C17:2 in this Figure?

I hope my comments will help improve your manuscript.

Author Response

find attached the response to comments

Round 2

Reviewer 4 Report

Comments and Suggestions for Authors

I agree with the improvement of the paper, but still some minor corrections are necessary.

- 2.3 section: specify concentration of standards used for UV spectra registration.

- 443-448 - chromatographic conditions shouldn't be repeated, as these conditions were not developed by the author, but were taken from Pharmacopoeia.